# Invariance in Policy Optimisation and Partial Identifiability in Reward Learning

## Abstract

Designing reward functions for complex, real-world tasks is challenging. *Reward learning* lets one instead *infer* reward functions from data. However, multiple reward functions often fit the data equally well, even in the infinite-data limit. Prior work often considers reward functions to be uniquely recoverable, by imposing additional assumptions on data sources. By contrast, we formally characterise the *partial identifiability* of popular data sources, including demonstrations and trajectory preferences, under multiple standard sets of assumptions. We analyse the impact of this partial identifiability on downstream tasks such as policy optimisation, including under shifts in environment dynamics. We unify our results in a framework for comparing data sources and downstream tasks by their invariances, with implications for the design and selection of data sources for reward learning.

## 1 Introduction

A wide range of problems can be represented as sequential decision-making tasks, where the goal is to maximise some numerical *reward* (Sutton & Barto, 2018). However, designing an appropriate *reward function* remains a challenge in complex real-world tasks (Amodei et al., 2016; Leike et al., 2018; Dulac-Arnold et al., 2019). *Reward learning* algorithms infer task reward functions from *data sources* such as expert demonstrations (Ng & Russell, 2000), preferences over trajectories (Christiano et al., 2017), and many others (Jeon et al., 2020). This approach has extended the applicability of sequential decision-making techniques to more complex tasks (e.g. Abbeel et al., 2010; Christiano et al., 2017; Singh et al., 2019; Stiennon et al., 2020).

Multiple reward functions are often consistent with the data source, even in the infinite-data limit. For most data sources, this fundamental *ambiguity* has been acknowledged, but its extent has not been characterised. In section 3, we formally characterise the *ambiguity* of several popular data sources including expert demonstrations and trajectory preferences. These infinite-data limits bound the information recoverable from finite data sets using any algorithm, so they are useful for evaluating algorithms relative to their limits, and data sources relative to each other.

Uniquely identifying a reward function is unnecessary when all plausible reward functions lead to the same downstream outcome in a given application, such as policy optimisation. Characterising this ambiguity *tolerance* for various applications allow us to evaluate the ambiguity of a data source *relative* to a given application. Learnt reward functions are often used for policy optimisation, for example via reinforcement learning (RL)[1]. In section 3 we formally characterise the ambiguity *tolerance* of policy optimisation under arbitrary dynamics.

Ambiguity and ambiguity tolerance are formally related. Both concern *invariances* – of data sources or downstream outcomes – to reward function *transformations*. Thus, our main contribution is to catalogue the invariances of various mathematical objects derived from the reward function. In section 4, we explore a *partial order* on these invariances, and its implications for the selection and evaluation of data sources, addressing an open problem in reward learning (Leike et al., 2018, §3.1).

---

[1]We focus on RL applications. Further applications arise in other fields, where reward functions are used in models to understand and predict the behaviour of humans, animals, and other systems (see, e.g., Schoemaker, 1982; Dennett, 1989; Rust, 1994; Howes et al., 2014; Peterson et al., 2021; Collins & Shenhav, 2021).

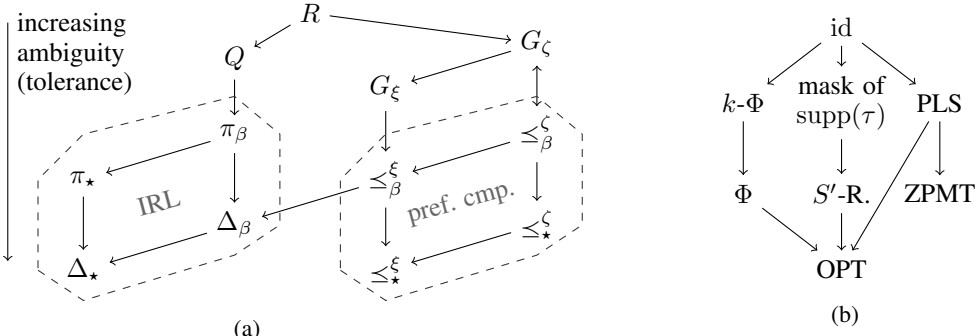

(a) (b)

Figure 1: (a) The *infinite-data ambiguity* of reward learning data sources, and the *ambiguity tolerance* of downstream applications of a learnt reward function, are both *invariances* of objects derived from reward functions (sections 1.2 and 3). These invariances have a *partial order* (section 4): here, $X \to Y$ means that $Y$ can be derived from $X$, or equivalently that $Y$ is at least as ambiguous as $X$. The objects are: the reward function itself ($R$); $Q$-functions ($Q$); Maximum Entropy ($\beta$) and supportive optimal policies ($\star$) and their induced trajectory distributions ($\pi_\beta$, $\Delta_\beta$ and $\pi_\star$, $\Delta_\star$); the return function restricted to partial and full trajectories ($G_\zeta$, $G_\xi$); Boltzmann-distributed ($\beta$) and noiseless ($\star$) comparisons between these trajectories ($\preceq_\beta^\zeta$, $\preceq_\beta^\xi$ and $\preceq_\star^\zeta$, $\preceq_\star^\xi$). (b) Several basic families of reward transformations form the basis for our main results (section 2). These transformations exist in a related hierarchy, within (shown here) and across tasks (section 4).

## 1.1 RELATED WORK

*Inverse reinforcement learning* (IRL; Russell, 1998) is the prototypical example of reward learning. IRL infers a reward function from the behavioural data of a task expert by inverting a model of the expert's *planning algorithm* (Armstrong & Mindermann, 2017; Shah et al., 2019). Existing work partially characterises the inherent ambiguity of behaviour for certain planning algorithms (Ng & Russell, 2000; Cao et al., 2021) and classes of tasks (Dvijotham & Todorov, 2010; Kim et al., 2021). We extend these results to more planning algorithms and arbitrary time-unbounded, stochastic tasks, using a more expressive space of reward functions that reveals novel ambiguity.

Reward learning models have been proposed for many other data sources (Jeon et al., 2020). A popular and effective data source is *preferences over behavioural trajectories* (Akrour et al., 2012; Christiano et al., 2017). Unlike for IRL, the ambiguity arising from these data sources has not been formally characterised. We contribute a formal characterisation of the ambiguity for central models of evaluative feedback including trajectory preferences.

Several studies have explored learning from expert behaviour *and* preferences (Ibarz et al., 2018; Palan et al., 2019; Bıyık et al., 2020; Koppol et al., 2020), or other multi-modal data sources (Tung et al., 2018; Jeon et al., 2020). One motivation is that different data sources may provide complementary reward information (Koppol et al., 2020), eliminating some ambiguity. Similarly, Amin et al. (2017) and Cao et al. (2021) observe reduced ambiguity by combining behavioural data across multiple *tasks*. Our partial order provides a general framework for understanding these results.

Computing an optimal behavioural policy is a primary application of learnt reward functions (Abbeel & Ng, 2004; Wirth et al., 2017). Ng et al. (1999) proved that *potential shaping transformations* always preserve the set of optimal policies, and so are always tolerable for this application. We extend this result, characterising the full set of transformations that preserve optimal policies in each task, including for additional policy optimisation techniques such as maximum entropy RL.

Ambiguity corresponds to the *partial identifiability* (Lewbel, 2019) of the reward function modelled as a latent parameter. The prevailing response to partial identifiability in reward learning has been to impose additional constraints or assumptions until the data identifies the reward function *uniquely* (or, at least, *sufficiently* for policy optimisation). Following Manski (1995; 2003) and Tamer (2010), we instead *describe* ambiguity *given* various constraints and assumptions. This gives practitioners results appropriate for their real data (and the ambiguity tolerance of their actual application).

IRL is related to *dynamic discrete choice* (Rust, 1994; Aguirregabiria & Mira, 2010), a problem where identifiability has been extensively studied (e.g., Aguirregabiria, 2005; Srisuma, 2015; Arcidiacono & Miller, 2020). We study a simpler setting with known tasks. IRL also relates to *preference elicitation* (Rothkopf & Dimitrakakis, 2011) and *inverse optimal control* (Ab Azar et al., 2020). Preferences over sequential trajectories are not typically considered as a data source in other fields.

## 1.2 PRELIMINARIES

We consider an idealised setting with finite, observable, infinite-horizon sequential decision-making environments, formalised as *Markov Decision Processes* (MDPs; Sutton & Barto, 2018, §3). An MDP is a tuple $(\mathcal{S}, \mathcal{A}, \tau, \mu_0, R, \gamma)$ where $\mathcal{S}$ and $\mathcal{A}$ are finite sets of environment *states* and agent *actions*; $\tau : \mathcal{S} \times \mathcal{A} \to \Delta(\mathcal{S})$ encodes the *transition distributions* governing the environment dynamics; $\mu_0 \in \Delta(\mathcal{S})$ is an initial state distribution; $R : \mathcal{S} \times \mathcal{A} \times \mathcal{S} \to \mathbb{R}$ is a deterministic *reward function*[2]; and $\gamma \in (0, 1)$ is a reward *discount rate*. We distinguish states in the support of $\mu_0$ as *initial states*, and states $s$ with $\tau(s|s, a) = 1$ and $R(s, a, s) = 0$ for all $a$ as *terminal states*.

We represent the *transition* from state $s$ to state $s'$ using action $a$ as the tuple $x = (s, a, s')$. We classify $(s, a, s')$ as *possible* in an MDP if $s'$ is in the support of $\tau(s, a)$, otherwise it is *impossible*. A *trajectory* is an infinite sequence of concatenate transitions $\xi = (s_0, a_0, s_1, a_1, s_2, \ldots)$, and a *trajectory fragment* of length $n$ is a finite sequence of $n$ concatenate transitions $\zeta = (s_0, a_0, s_1, \ldots, a_{n-1}, s_n)$. A trajectory or fragment is *possible* if all of its transitions are possible, and is *impossible* otherwise. A trajectory or fragment is *initial* if its first state is initial. A state or transition is *reachable* if it is part of some possible and initial trajectory.

Given an MDP, we define the *return function $G$* as the cumulative discounted reward of entire trajectories and trajectory fragments: $G(\zeta) = \sum_{t=0}^{n-1} \gamma^t R(s_t, a_t, s_{t+1})$ for a trajectory fragment $\zeta$ of length $n$, and similarly for trajectories. We primarily consider this return function with various *restricted domains*, such as only *possible* or *initial* trajectories or trajectory fragments.

A *policy* $\pi : \mathcal{S} \to \Delta(\mathcal{A})$ encodes an agent's behaviour as a state-conditional action distribution. Together with an MDP's transition distribution $\tau$, a policy $\pi$ induces a distribution of trajectories starting from each state. We denote such a trajectory starting from $s$ with the random variable $\Xi_s$, and its remaining components with random variables $A_0, S_1, A_1, S_2$, and so on.

Given an MDP and a policy $\pi$, and the *value function* encodes the expected return from states, $V_\pi(s) = \mathbb{E}_{\Xi_s \sim \pi, \tau}\big[G(\Xi_s)\big]$; and the *Q-function* of $\pi$ encodes the expected return given an initial action, $Q_\pi(s, a) = \mathbb{E}_{\Xi_s \sim \pi, \tau}\big[G(\Xi_s) \mid A_0 = a\big]$. $Q_\pi$ and $V_\pi$ satisfy a *Bellman equation*:

$$Q_\pi(s, a) = \mathbb{E}_{S' \sim \tau(s,a)}\big[R(s, a, S') + \gamma V_\pi(S')\big], \qquad V_\pi(s) = \mathbb{E}_{A \sim \pi(s)}\big[Q_\pi(s, A)\big], \qquad (1)$$

for all $s \in \mathcal{S}$ and $a \in \mathcal{A}$. Their difference, $A_\pi(s, a) = Q_\pi(s, a) - V_\pi(s)$, is the *advantage function*.

We further define a *policy evaluation function*, $\mathcal{J}$, encoding the expected return from following a particular policy in an MDP, $\mathcal{J}(\pi) = \mathbb{E}_{S_0 \sim \mu_0}\big[V_\pi(S_0)\big]$. $\mathcal{J}$ induces an order over policies. A policy maximising $\mathcal{J}$ is an *optimal policy*, denoted $\pi_\star$. Similarly, $Q_\star$, $V_\star$, and $A_\star$ denote the $Q$-, value, and advantage functions of an optimal policy. Since $\mathcal{J}$ may be multimodal, we often discuss the *set* of optimal policies. However, $Q_\star$, $V_\star$, and $A_\star$ are unique.

Moreover, we consider several policies resulting from alternative *planning algorithms*. Given a base policy $\pi_0$, and an *inverse temperature* parameter $\beta > 0$, we define the *Boltzmann policy* with respect to $\pi_0$, denoted $\pi_\beta^{\pi_0}$, using the softmax function:

$$\pi_\beta^{\pi_0}(a|s) = \frac{\exp\big(\beta A_{\pi_0}(s, a)\big)}{\sum_{a' \in \mathcal{A}} \exp\big(\beta A_{\pi_0}(s, a')\big)}. \qquad (2)$$

The *Boltzmann-rational* policy, $\pi_\beta^\star$, is the Boltzmann policy with respect to optimal policies, as used for IRL by Ramachandran & Amir (2007). The popular *Maximum Entropy* policy, $\pi_\beta$, is defined using a modified policy evaluation function with an entropy regularisation term (Haarnoja et al., 2017), and solves the recurrence $\pi_\beta = \pi_\beta^{\pi_\beta}$ (i.e., $\pi_\beta$ is a Boltzmann policy with respect to itself).

---

[2]Notably, we consider deterministic reward functions that may depend on a transition's successor state. Alternative spaces of reward functions are often considered (such as functions from $\mathcal{S}$ or $\mathcal{S} \times \mathcal{A}$, or distributions). The chosen space has straightforward consequences for invariances, which we discuss in appendix C.

## 2 Reward Function Transformations

In this section, we discuss how *invariance* to reward function *transformations* relates to infinite-data ambiguity in reward learning and ambiguity tolerance in applications.

**Definition 2.1** (Transformations and invariances). A *transformation* is a map between reward functions. The *invariances* of an object $X$ derived from reward $R$ via function $f$ are all transformations $t$ that preserve $f$: $X = f(R) = f(t(R))$ for all $R$. We say that $X$ *determines $R$ up to* its invariances.

A set of transformations carves out a partition of the space of reward functions by grouping together those reward functions reachable from one another using the transformations. The partition carved out by the invariances of an object is the *equivalence kernel* of the object's derivation function – grouping the reward functions from which identical objects are derived into partition blocks.

Given a reward learning data source, consider the object encoding the information available from the data source in the infinite-data limit (Lewbel, 2019, §3.1). The invariances of this object represent the *infinite-data ambiguity* of the data source – it is impossible to recover the reward function beyond the corresponding partition block, as the remaining functions imply indistinguishable data.

Similarly, consider a downstream application of learnt reward functions involving the computation of an object. The object's invariances capture the *ambiguity tolerance* of this computation, as by definition all reward functions in each block of the corresponding partition lead to identical outcomes.

### 2.1 Fundamental Reward Transformations

We introduce several fundamental sets of reward function transformations, forming a basis for the invariances we study in section 3. Before considering several novel transformations, we recall *potential shaping*, introduced by Ng et al. (1999) and widely known to preserve optimal policies in all MDPs. We further distinguish special potential shaping transformations with constant potential over an MDP's initial states. We explore some properties of potential shaping in appendix A.

**Definition 2.2** (Potential Shaping). A *potential function* is a function $\Phi : \mathcal{S} \to \mathbb{R}$, where $\Phi(s) = 0$ if $s$ is a terminal state. If $\Phi(s) = k$ for all initial states then we say that $\Phi$ is *$k$-initial*. Let $R$ and $R'$ be reward functions. Given a discount $\gamma$, we say $R'$ is produced by *($k$-initial) potential shaping* of $R$ if $R'(s, a, s') = R(s, a, s') + \gamma \cdot \Phi(s') - \Phi(s)$ for some ($k$-initial) potential function $\Phi$.

**Definition 2.3** ($S'$-Redistribution). Let $R$ and $R'$ be reward functions. Given transition dynamics $\tau$, say $R'$ is produced by *$S'$-redistribution* of $R$ if $\mathbb{E}_{S' \sim \tau(s,a)}\big[R(s, a, S')\big] = \mathbb{E}_{S' \sim \tau(s,a)}\big[R'(s, a, S')\big]$.

$S'$-redistribution allows changing $R$ arbitrarily for impossible transitions. Moreover, if at least two states $s'_1$, $s'_2$ are in the support of $\tau(s, a)$ then $S'$-redistribution lets us increase $R(s, a, s'_1)$ and decrease $R(s, a, s'_2)$ by a proportionate amount. Note that $S'$-redistribution depends crucially on the reward function's dependence on the successor state. This set of transformations collapses to the identity for simpler spaces of reward functions, as we explore in appendix C.

**Definition 2.4** (Monotonic Transformations). Let $R$ and $R'$ be reward functions. Say $R'$ is produced by a *zero-preserving monotonic transformation* of $R$ if for all pairs of transitions $x, x' \in \mathcal{S} \times \mathcal{A} \times \mathcal{S}$, $R(x) \leqslant R(x')$ if and only if $R'(x) \leqslant R'(x')$, and $R(x) = 0$ if and only if $R'(x) = 0$. Moreover, say $R'$ is produced by *positive linear scaling* of $R$ if $R' = c \cdot R$ for some positive constant $c$.

A zero-preserving monotonic transformation is simply a monotonic transformation that maps zero to itself. Positive linear scaling is a special case.

**Definition 2.5** (Optimality-Preserving Transformation). Let $R$ and $R'$ be reward functions. Given a function $\mathcal{O} : \mathcal{S} \to \mathcal{P}(\mathcal{A}) - \{\varnothing\}$, transition dynamics $\tau$, and discount rate $\gamma$, we say $R'$ is produced from $R$ by an *optimality-preserving transformation* with $\mathcal{O}$ if there is a function $\Psi : \mathcal{S} \to \mathbb{R}$ such that $\mathbb{E}_{S' \sim \tau(s,a)}\big[R'(s, a, S') + \gamma \cdot \Psi(S')\big] \leqslant \Psi(s)$ for all $s, a$, with equality if and only if $a \in \mathcal{O}(s)$.

This transformation gives the reward functions with optimal actions from $\mathcal{O}$ ($\Psi$ determines the new value function). Note that in practice an implicit dependence on $R$ is introduced through the definition of $\mathcal{O}$. Also, we can reach arbitrary $R'$ if $\mathcal{O}$ is unconstrained (in practice, we constrain $\mathcal{O}$).

Finally, we consider transformations allowing the reward to vary freely for a given set of transitions.

**Definition 2.6** (Masking). Let $R$ and $R'$ be reward functions. Given a transition set $\mathcal{X} \subseteq \mathcal{S} \times \mathcal{A} \times \mathcal{S}$, say $R'$ is produced by a *mask of $\mathcal{X}$* from $R$ if $R(x) = R'(x)$ for all $x \notin \mathcal{X}$.

## 3 INVARIANCES OF REWARD-RELATED OBJECTS

In this section we catalogue the *invariances* of various central objects derived from reward functions, including expert trajectory distributions, the ranking of trajectories induced by the return function, and the set of optimal policies. Some of these objects correspond to the information available in the infinite-data limit of a reward learning data source, while others correspond to the outcome of a downstream application.

If an object $X$ can be derived from another object $Y$ without further reference to the reward function, then $X$ inherits $Y$'s invariances. For example, the optimal $Q$-function's invariances are inherited by various expert policies. Accordingly, we organise this section by incrementally deriving our objects of interest starting from the reward function, cataloguing the invariances introduced in each step. This also mirrors the structure of figure 1a. We defer all proofs until appendix B.

### 3.1 INVARIANCES OF EXPERT BEHAVIOUR

Inverse reinforcement learning (IRL) algorithms infer a task's reward function from the behaviour of task experts. Formally, this behaviour is represented as an expert's *policy* or a sample of *trajectories*.

To characterise the corresponding invariances, we begin with $Q$-functions – instrumental to deriving many policies. $Q$-functions are invariant to $S'$-redistribution since they are defined as an expectation over the successor state $S'$, We show that this is the only invariance for $Q$-functions.

**Theorem 3.1.** *Given an MDP and a policy $\pi$, the $Q$-function for $\pi$, $Q_\pi$, determines $R$ up to $S'$-redistribution. The optimal $Q$-function, $Q_\star$, has precisely the same invariances.*

This invariance is inherited by any object that can be derived from a $Q$-function. However, note that $S'$-redistribution vanishes in simpler spaces of reward functions, as we explore in appendix C.

We now turn to policies derived using various planning algorithms. These policies are instrumental in constructing the trajectories studied in IRL. For example, Ramachandran & Amir (2007) and Ziebart et al. (2008) assume that expert behaviour is drawn from a *Boltzmann-rational policy*, and Ziebart et al. (2010) assume a *Maximum Entropy policy* We catalogue the invariances of arbitrary *Boltzmann policies*, of which these other policies are special cases. As these policies can be derived from $Q$-functions, they inherit invariance to $S'$-redistribution. We show they are also invariant to potential shaping, but not to any other transformations.

**Theorem 3.2.** *Given an MDP, an inverse temperature parameter $\beta$, and a base policy $\pi_0$, the Boltzmann policy $\pi_\beta^{\pi_0}$ determines $R$ up to $S'$-redistribution and potential shaping. The Boltzmann-rational policy, $\pi_\beta^\star$, and the Maximum Entropy policy, $\pi_\beta$, have precisely the same invariances.*

By contrast, Ng & Russell (2000) and Abbeel & Ng (2004) assume that experts follow an *optimal policy*. Optimal policies inherit $S'$-redistribution invariance from the optimal $Q$-function, and are also known to be invariant to potential shaping (Ng et al., 1999). Under an additional assumption that a given policy is *maximally supportive*, in that it takes all optimal actions with positive probability, we show that *these invariances and any additional invariances* are captured in a class of *optimality-preserving transformations* (Definition 2.5) based on the set of optimal actions in each state.

**Theorem 3.3.** *Given an MDP, let $\mathcal{O}(s) = \arg\max_a A_\star(s, a)$. A maximally supportive optimal policy determines $R$ up to optimality-preserving transformations with $\mathcal{O}$.*

Additional invariances arise if we consider optimal policies that may lack support for optimal actions. As a well-known example, the zero-reward is consistent with any policy in this sense.

In the infinite-data limit, a data source of trajectories sampled from a policy reveals the *distribution* of trajectories induced by the policy, and therefore *the policy itself* for all states reachable via its supported actions. A Boltzmann policy supports all actions, so in the infinite data limit, samples of trajectories determine the policy for all *reachable* states. We show this introduces invariance precisely to changes in the reward of unreachable transitions.

**Theorem 3.4.** *Given an MDP, an inverse temperature parameter $\beta$, and a base policy $\pi_0$, the distribution of trajectories induced by the Boltzmann policy $\pi_\beta^{\pi_0}$ from all initial states determines $R$ up to $S'$-redistribution, potential shaping, and a mask of* unreachable *transitions. The distributions of trajectories induced by the Boltzmann-rational policy, $\pi_\beta^\star$, and the Maximum Entropy policy, $\pi_\beta$, from all initial states, have precisely the same invariances.*

Similarly, trajectories sampled from an optimal policy reveal the policy in those states that its actions reach. This again introduces additional invariance to transformations of reward in other states.

**Theorem 3.5.** *Given an MDP, consider the distribution of trajectories induced by a maximally supportive optimal policy. Let $\mathfrak{S}$ be the set of states in supported trajectories. Let $\mathfrak{D}$ be the set of functions $\mathcal{O}$ defined on $\mathcal{S}$ such that $\mathcal{O}(s) = \arg\max_a A_\star(s, a)$ for all $s \in \mathfrak{S}$. The induced distribution of trajectories determines $R$ up to optimality-preserving transformations with $\mathcal{O} \in \mathfrak{D}$.*

Note that a mask of the complement of $\mathfrak{S}$ is *not* permitted. However, the fact that $\mathcal{O}$ is unconstrained outside $\mathfrak{S}$ leaves reward effectively unconstrained in those states, except that the reward of transitions out of $\mathfrak{S}$ may have to "compensate" for the value of their successor states, to prevent new actions that lead out of $\mathfrak{S}$ from becoming optimal.

## 3.2 Invariances of Trajectory Evaluation

The return function, capturing the reward accumulated over a trajectory, is instrumental in deriving data for evaluative feedback such as reward labels and trajectory preference comparisons. We consider the invariances of the return function for various restricted domains.

**Theorem 3.6.** *Given an MDP, the return function restricted to possible trajectory fragments, $G_\zeta$, determines $R$ up to a mask of* impossible *transitions;*

**Theorem 3.7.** *Given an MDP, the return function restricted to possible and initial trajectories, $G_\xi$, determines $R$ up to zero-initial potential shaping and a mask of* unreachable *transitions.*

The limited invariance of the return of fragments arises because this restricted domain still includes individual (possible) transitions. Additional invariances will arise from additional restrictions, such as a minimum or maximum fragment length, or a restriction to initial trajectory fragments.

Pairwise comparisons between trajectories are studied as a data source for reward learning (Akrour et al., 2012; Christiano et al., 2017). It is common to model the comparisons as based on the return of trajectories, but with accompanying *decision noise* following a *Boltzmann distribution*. Under this assumption, in the limit of infinite noisy comparisons for each pair of trajectories, the data source reveals the Boltzmann distributions. Boltzmann noise encodes relative cardinal information about the return of trajectories and fragments, so little invariance is introduced.

Formally, given an MDP and an inverse temperature parameter $\beta > 0$, let $\preceq_\beta^\zeta$ be a distribution over each pair of *possible* trajectory fragments, $\zeta_1, \zeta_2$, such that

$$\mathbb{P}(\zeta_1 \preceq_\beta^\zeta \zeta_2) = \frac{\exp(\beta G(\zeta_2))}{\exp(\beta G(\zeta_1)) + \exp(\beta G(\zeta_2))},$$

and let $\preceq_\beta^\xi$ be the analogous distribution over each pair of *possible* and *initial* trajectories.

**Theorem 3.8.** *Given an MDP, the distribution of comparisons of possible trajectory fragments, $\preceq_\beta^\zeta$, determines $R$ up to a mask of* impossible *transitions.*

**Theorem 3.9.** *Given an MDP, the distribution of comparisons of possible and initial trajectories, $\preceq_\beta^\xi$, determines $R$ up to $k$-initial potential shaping and a mask of* unreachable *transitions.*

The limited invariance of Boltzmann comparisons of fragments arises from the very flexible comparisons permitted, including, for example, comparisons between individual transitions and empty trajectories. Additional invariances will arise from additional restrictions, such as permitting comparisons only between fragments of a fixed length. Moreover, it is worth reiterating that these invariances rely heavily on the precise structure of the decision noise revealing cardinal information in the infinite-data limit.

It is also possible to model trajectory comparisons as noiseless comparisons based on the return. The infinite data limit then corresponds to the order induced by the return functions. Formally, define the *noiseless order of possible trajectory fragments* as a relation, $\preceq_\star^\zeta$, on possible trajectory fragments:

$$\zeta_1 \preceq_\star^\zeta \zeta_2 \Leftrightarrow G(\zeta_1) \leqslant G(\zeta_2).$$

Similarly, define the *noiseless order of possible and initial trajectories* as the analogous relation, $\preceq_\star^\xi$, for pairs of *possible* and *initial* trajectories. These relations omit cardinal information about pairwise comparisons, and so invariances to certain monotonic transformations are introduced. The precise monotonic invariances depend on the MDP (for example, see the proof in appendix B.3).

**Theorem 3.10.** *We have the following bounds on the invariances of the noiseless order of possible trajectory fragments, $\preceq_\star^\zeta$. In all MDPs:*

(1) *$\preceq_\star^\zeta$ is invariant to positive linear scaling and a mask of impossible transitions; and*

(2) *$\preceq_\star^\zeta$ is not invariant to transformations other than zero-preserving monotonic transformations or masks of impossible transitions.*

*Moreover, there exist MDPs attaining each of these bounds.*

We give a lower bound on the invariances of the noiseless order of possible and initial trajectories, $\preceq_\star^\xi$. Since $\preceq_\star^\xi$ can be derived from $\preceq_\beta^\xi$, it inherits the latter's invariances. Moreover, like $\preceq_\star^\zeta$, $\preceq_\star^\xi$ is always invariant to positive linear scaling.

**Theorem 3.11.** *Given an MDP, the noiseless order of possible and initial trajectories, $\preceq_\star^\xi$, is invariant to $k$-initial potential shaping, positive linear scaling, and a mask of unreachable transitions.*

### 3.3 INVARIANCES OF POLICY OPTIMISATION

The primary application of learnt reward functions is to compute optimal policies, using techniques such as RL. Policy optimisation procedures typically compute a single optimal policy. However, in terms of invariances, one may desire to preserve the whole *set* of optimal policies, so as not to tolerate any sub-optimal policies *becoming* optimal through a reward transformation.

The set of optimal policies inherits $S'$-redistribution invariance from the optimal $Q$-function, and is also known to be invariant to potential shaping (Ng et al., 1999). In fact, because a maximally supportive optimal policy can be derived from the set of optimal policies *and vice versa*, the set shares the same invariances as a maximally supportive optimal policy (Theorem 3.3).

**Theorem 3.12.** *Given an MDP, let $\mathcal{O}(s) = \arg\max_a A_\star(s, a)$. Then the set of optimal policies determines $R$ up to optimality-preserving transformations with $\mathcal{O}$.*

Moreover, if one uses an algorithm not guaranteed to find a globally optimal policy, one may desire to preserve the entire *order* induced on the space of policies by the policy evaluation function, rather than just the set of maximising policies. Future work could investigate the invariances of the ordinal information in the policy evaluation function. Note that since the set of optimal policies can be derived from this order, the order has at most the invariances of the set of optimal policies.

Finally, we sketch some bounds on the invariances of the set of optimal policies across *all MDPs*. Potential shaping and linear scaling preserve optimal policies in each MDP, and hence in all MDPs. $S'$-redistribution and optimality-preserving transformations for a given MDP might not. Moreover, Theorem 3.12 implies that any transformation that is not an optimality-preserving transformation in a given MDP cannot preserve optimal policies in that MDP, let alone all MDPs.

## 4 IMPLICATIONS FOR REWARD LEARNING

So far we have catalogued the *invariances* of transformations to the reward function of various reward function derived objects. These invariances characterise the *infinite-data ambiguity* of several reward learning data sources, and the *ambiguity tolerance* of policy optimisation. In this section, we discuss the implications for the practical evaluation of reward learning data sources.

We begin by defining a mathematical framework for comparisons between data sources and applications in terms of their ambiguity. The characterisation of ambiguity and tolerance as invariances to reward transformations suggests a natural *partial order* on data sources and applications. Recall that the invariances of an object correspond to a *partition* of the space of reward functions (section 2). We lift the *refinement relation* for partitions (Aigner, 1979, §I.2.B) to data sources and applications as follows.

**Definition 4.1** (Ambiguity refinement)**.** Consider two reward learning data sources (or applications), $X$ and $Y$. Let $\Pi_X$ and $\Pi_Y$ be the partitions of the space of reward functions corresponding to their respective invariances (definition 2.1). If $\Pi_X$ is a partition refinement of $\Pi_Y$, we write $X \preceq Y$, and we say $X$ is *no more ambiguous* than $Y$ (or $X$ is *tolerable for* application $Y$). If $X \preceq Y$ but not $Y \preceq X$, then we write $X \prec Y$ and say $X$ is (strictly) *less ambiguous* than $Y$.

Given two data sources $X$ and $Y$, $X \preceq Y$ corresponds to $X$ conflating no additional reward functions compared to $Y$ in the infinite-data limit. This is the sense in which we say $X$ is *no more ambiguous* than $Y$. Moreover, given a downstream application $Z$, $X \preceq Z$ is precisely the condition of $Z$ tolerating the infinite-data ambiguity of data source $X$: $X \preceq Z$ if and only if the reward functions conflated by $X$ in the infinite-data limit all lead to the same outcome in $Z$.

Of our fundamental reward functions transformations, there are several clear instances of ambiguity refinement in a given MDP, as summarised in figure 1b. Invariance to $k$-initial potential shaping ($k$-$\Phi$) corresponds to less ambiguity than general potential shaping ($\Phi$). Likewise positive linear scaling (PLS) is less ambiguous than zero-preserving monotonic transformations (ZPMT), and a mask of impossible transitions is less ambiguous than $S'$-redistribution ($S'$-R). All of these transformations other than zero-preserving monotonic transformations are less ambiguous than the optimality-preserving transformations we have encountered.

More concretely, we can compare the ambiguity of specific data sources. Some of these comparisons are indicated in figure 1a. For example, the ambiguity tolerance of the set of optimal policies is a class of optimality-preserving transformations. Each of the data sources that are less ambiguous than this tolerance are sufficient for policy optimisation.

Notably, this excludes noiseless comparisons between trajectory fragments in some MDPs. Specifically, policy optimisation does not, in general, tolerate zero-preserving monotonic transformations (ZPMT), while noiseless comparisons are invariant to this transformation in some MDPs (Theorem 3.10). Policy optimisation also does not tolerate data sources based on possible and initial trajectories, which are invariant to a mask of unreachable transitions. However, these sources are tolerable if the application only requires optimal behaviour in reachable states.

Moreover, we can compare data sources drawn from one MDP to applications in *another* MDP, such as under a shift in transition dynamics or initial state distribution. This captures the common sim-to-real setting where learning takes place in a simulated or otherwise restricted environment that differs from the final deployment environment. The simplest transformations to consider are masks of *possible* or *reachable* transitions. These are parametrised by transition dynamics. In general, the ambiguity corresponding to a mask of $\mathcal{X}$ is less than for a mask of $\mathcal{X}' \supset \mathcal{X}$. For example, if the new dynamics supports transitions that were previously impossible, then sources with invariance to a mask from the original MDP may not be tolerable for applications in the new MDP.

A similar results holds for $S'$-redistribution, which involves an expectation over MDP dynamics. As an extreme example, we prove that when the transition dynamics are changed for every state and action, $S'$-redistribution under the original dynamics permits an arbitrary $Q$-function under the new dynamics. Naturally, data sources derived from $Q$-functions may also be affected by shifts in dynamics. Note that this strong result relies on the formulation of rewards as depending on the successor-state (cf. appendix C).

**Theorem 4.1.** *Consider an MDP $(\mathcal{S}, \mathcal{A}, \tau, \mu_0, R, \gamma)$, a policy $\pi$, and alternative transition dynamics $\tau'$ with $\tau(s, a) \neq \tau'(s, a)$ for all $s \in \mathcal{S}, a \in \mathcal{A}$. Given a function $Q' : \mathcal{S} \times \mathcal{A} \to \mathbb{R}$, there exists a reward function $R'$, produced from $R$ by $S'$-redistribution under $\tau$, such that $Q'$ is the $Q$-function for $\pi$ under $R'$ and $\tau'$.*

Ambiguity refinement is a *partial* order, and some data sources are indeed *incomparable*. In consolation, we observe that such incomparable ambiguity is *complementary* ambiguity, in that by combining the associated data sources, we reduce overall ambiguity about the latent reward.

**Theorem 4.2.** *Given data sources $X$ and $Y$, let $(X, Y)$ denote the combined data source formed from $X$ and $Y$. If $X$ and $Y$ are incomparable, then $(X, Y) \prec X$ and $(X, Y) \prec Y$.*

This perspective highlights promising directions for the design of reward learning data sources. In particular, this suggests developing reward learning algorithms for mixtures of data sources with *complementary* ambiguity. Unfortunately, most popular data sources actually appear to have similar kinds of ambiguity given one MDP. However, ambiguity could be reduced by incorporating data from *multiple MDPs*, along the lines of Amin et al. (2017) and Cao et al. (2021).

## 5 LIMITATIONS AND FUTURE WORK

Our results give an upper bound on the amount of information that can be extracted from a given data source. However, in practice, these bounds may not be reached. In particular, our results are for the limit of infinite data. But in practice data sets are finite and, when data collection is expensive, may be fairly small. An important direction for future work is to characterise how much information is contained in data sets of varying sizes and data sources. This would enable practitioners to determine the most sample efficient data source for a fixed data collection budget.

Furthermore, our results rely on the data being generated according to the process assumed by the reward learning algorithm. However, most popular approaches are a poor fit for human data (Orsini et al., 2021). For example, human demonstrations are rarely perfectly optimal or Boltzmann-rational. Moreover, there is often a trade-off between how informative a data source is and how easy it is for a user to provide data. As an extreme example, a user directly specifying the target reward function is maximally informative – *if* users could complete such a task correctly. We expect the maximum informativeness of a data source to be a useful metric, but it should be considered alongside the cost and tractability of collecting different kinds of data.

## 6 CONCLUSION

Substantial effort has been invested to develop reward learning algorithms for a variety of data sources. A fundamental question to ask is how effective are these algorithms relative to an optimal algorithm for that data source? Our results characterise the information available in different data sources, enabling algorithms to be compared to this theoretical upper bound.

Moreover, our framework enables direct comparisons between different data sources. We find that some data sources are strictly less informative than others, such as noiseless preference comparisons vs. return labels. By contrast, others are incomparable and have complementary ambiguity, such as $Q$-values (invariant to $S'$-redistribution but not potential shaping) vs. episode return $G_\xi$ (invariant to some potential shaping, but not to $S'$-redistribution).

In particular, we have characterised the invariances of various reward-related objects to transformations such as potential shaping. We have shown that these objects form a partial order under *ambiguity refinement*. These results, summarised in figure 1, allow us to predict the ambiguity of data sources generated from these objects. While practitioners could simply collect data from the least ambiguous source, this might be very expensive. Our framework also identifies the ambiguity tolerance of downstream applications (such as policy optimisation) that need to compute these objects. This enables practitioners to identify reward learning data sources with low ambiguity in the areas their application is sensitive to, enabling higher performance without unnecessary costs.

### ETHICS STATEMENT

It is important that AI systems are aligned with the interests of users and other stakeholders. In open-ended problems, directly specifying how the AI system should behave is intractable. Prior work has identified reward learning as an essential building block for AI systems that can cooperate with humans (Dafoe et al., 2020, §4.1.3), especially for powerful AI systems (Bostrom, 2014, chapter 12). We hope that our work provides greater clarity on both the limits and potential of various reward learning data sources. However, given the importance of the domain, we should stress that our work provides only one useful angle by which to evaluate data sources. In particular, we do not consider sample efficiency, robustness to misspecification, or the cost of data collection.

Moreover, even if a theoretically optimal and practically robust reward learning algorithm were to be developed, there would still remain important normative questions. In particular, *what* kinds of values we are aligning the AI system to – stated preferences, revealed preferences, instructions, or something else (Gabriel, 2020)? Additionally, it is important that all relevant stakeholders are able to provide input into the system. This may constrain the kinds of data we can collect. For example, while only task experts might be able to provide demonstrations, a wider variety of stakeholders might be able to provide preference comparisons. While a thorough evaluation of these considerations are beyond the scope of this paper, we would encourage practitioners to evaluate reward learning data sources holistically, including but not wholly relying on our results.

REPRODUCIBILITY STATEMENT

Our results are all theoretical in nature. We introduce notation and other background material in sections 1.2 and 2. Necessary assumptions are listed there and in each theorem statement. Proofs for some fundamental lemmas are provided in appendix A and all other proofs are in appendix B.

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

# A    PROPERTIES OF FUNDAMENTAL REWARD TRANSFORMATIONS

We begin with some supporting results concerning the basic reward transformations, used in section 3 to characterise the invariances of various objects derived from the reward function.

The following result captures how potential shaping affects various reward-related functions.

**Lemma A.1.** *Consider $M$ and $M'$, two MDPs differing only in their reward functions, respectively $R$ and $R'$. Denote the return, $Q$-, value, and policy evaluation functions of $M'$ by $G'$, $Q'_\pi$, $V'_\pi$, and $\mathcal{J}'$. If $R'$ is produced by potential shaping of $R$ with a potential function $\Phi$, then:*

*(1) for a trajectory fragment $\zeta = (s_0, a_0, s_1, \ldots, s_n)$, $G'(\zeta) = G(\zeta) + \gamma^n \Phi(s_n) - \Phi(s_0)$;*

*(2) for a trajectory $\xi = (s_0, a_0, \ldots)$, $G'(\xi) = G(\xi) - \Phi(s_0)$;*

*(3) for a state $s \in \mathcal{S}$, and action $a \in \mathcal{A}$, $Q'_\pi(s, a) = Q_\pi(s, a) - \Phi(s)$;*

*(4) for a state $s \in \mathcal{S}$, $V'_\pi(s) = V_\pi(s) - \Phi(s)$; and*

*(5) for a policy $\pi$, $\mathcal{J}'(\pi) = \mathcal{J}(\pi) - \mathbb{E}_{S_0 \sim \mu_0}\big[\Phi(S_0)\big]$.*

*Proof.* (1) is given by straightforward induction on the length of $\zeta$. For (2), take the limit as the length of a prefix goes to infinity, whereupon $\gamma^n \Phi(s_n)$ goes to zero ($\gamma < 1$ by definition, and $\Phi(s_n)$ is bounded since its domain is finite). (3) and (4) were proved for optimal policies by Ng et al. (1999), and they also observed that the extension to arbitrary policies is straightforward (following immediately from (2), for example). (5) is immediate from (4). □

The following results explain how $k$-initial potential shaping and linear scaling of $R$ correspond to affine transformations of $G$.

**Lemma A.2.** *Let $(\mathcal{S}, \mathcal{A}, \tau, \mu_0, R, \gamma)$ be an MDP, $R'$ a reward function, and $k \in \mathbb{R}$ a constant. Then we have that $G'(\xi) = G(\xi) - k$ for all possible and initial trajectories $\xi$, if and only if $R'$ is produced from $R$ by $k$-initial potential shaping and a mask of unreachable transitions.*

*Proof.* The converse follows from Lemma A.1 and that varying the reward for unreachable transitions does not affect the return of any possible, initial trajectories, by definition.

In the forward direction, consider an arbitrary reachable state $s \in \mathcal{S}$. Let $\zeta$ be a possible, initial, length $n$ trajectory fragment ending in $s$ (at least one exists, since $s$ is reachable). Let $\xi_s$ be some possible trajectory starting in $s$, and let $\Delta_{\xi_s} = G(\xi_s) - G'(\xi_s)$. Let $\zeta + \xi_s$ denote the concatenation of $\zeta$ and $\xi_s$. Since $\zeta + \xi_s$ is an initial trajectory, we have by assumption $G(\zeta + \xi_s) - G'(\zeta + \xi_s) = k$. Moreover, note that $G(\zeta + \xi_s) = G(\zeta) + \gamma^n G(\xi_s)$, (and likewise for $G'$), by definition of return. Then (recalling that we have defined $\gamma > 0$),

$$
\begin{aligned}
\Delta_{\xi_s} &= G(\xi_s) - G'(\xi_s) \\
&= \frac{G(\zeta + \xi_s) - G(\zeta)}{\gamma^n} - \frac{G'(\zeta + \xi_s) - G'(\zeta)}{\gamma^n} \\
&= \frac{k - G(\zeta) + G'(\zeta)}{\gamma^n} .
\end{aligned}
$$

Therefore $\Delta_{\xi_s}$ is independent of $\xi_s$, except for possible dependence on $\xi_s$'s starting state $s$. We associate a unique $P(s) = \Delta_{\xi_s}$ with each reachable $s$.

Now consider a reachable transition $(s, a, s')$. Further straightforward algebraic manipulation shows that $R'(s, a, s') = R(s, a, s') + \gamma \cdot P(s') - P(s)$. Moreover, from the definition of terminal states we must have that $P(s) = \Delta_{\xi_s} = 0$ for terminal $s$, and by assumption we have that $P(s) = \Delta_{\xi_s} = k$ if $s$ is initial. So $R'$ is given by $k$-initial potential shaping of $R$ with $\Phi(s) = P(s)$ for reachable transitions. Any variation in reward for unreachable transitions can be accounted for by the mask. □

**Lemma A.3.** *Let $(\mathcal{S}, \mathcal{A}, \tau, \mu_0, R, \gamma)$ be an MDP, $R'$ a reward function, and $c \in \mathbb{R}$ a constant. Then $G'(\xi) = c \cdot G(\xi)$ for all possible initial trajectories $\xi$, if and only if $R'$ is produced from $R$ by zero-initial potential shaping, linear scaling by a factor of $c$, and a mask of all unreachable transitions.*

*Proof.* From Lemma A.1, we have that 0-initial potential shaping leaves $G$ unchanged. Likewise, a mask of unreachable transitions does not affect $G$ for possible initial trajectories. Moreover, it is straightforward that linear scaling of $R$ by a factor of $c$ leads to a linear scaling of $G$ by the same factor of $c$. Hence the converse is established.

For the forward direction, consider an arbitrary reachable state $s \in \mathcal{S}$. Let $\xi_s$ be some possible trajectory starting in $s$, and let $\Delta_{\xi_s} = c \cdot G(\xi_s) - G'(\xi_s)$. Let $\zeta$ be a possible, initial, length $n$ trajectory fragment ending in $s$. Since the concatenation $\zeta + \xi_s$ is a possible initial trajectory, we have that

$$G'(\zeta + \xi_s) = c \cdot G(\zeta + \xi_s)$$
$$\rightarrow \qquad G'(\zeta) + \gamma^n G'(\xi_s) = c \cdot (G(\zeta) + \gamma^n G(\xi_s))$$
$$\rightarrow \qquad G'(\zeta) = c \cdot G(\zeta) + \gamma^n (c \cdot G(\xi_s) - G'(\xi_s))$$
$$= c \cdot G(\zeta) + \gamma^n \Delta_{\xi_s}$$
$$\rightarrow \qquad \Delta_{\xi_s} = \frac{G'(\zeta) - c \cdot G(\zeta)}{\gamma^n}$$

Therefore $\Delta_{\xi_s}$ is independent of $\xi_s$, except for possible dependence on $\xi_s$'s starting state $s$. We associate a unique $P(s) = \Delta_{\xi_s}$ with each reachable $s$.

Now consider a reachable transition $(s, a, s')$. Further straightforward algebraic manipulation shows that $R'(s, a, s') = c \cdot R(s, a, s') + \gamma \cdot P(s') - P(s)$. Moreover, from the definition of terminal states we must have that $P(s) = \Delta_s = 0$ for terminal $s$, and by assumption we have that $P(s) = 0$ if $s$ is initial. Thus $R'$ is given by first potential shaping $R$ with $\Phi(s) = \frac{1}{c} P(s)$, and then linear scaling with $c$. $\qquad \square$

# B PROOFS

We provide proofs for the theoretical results presented in the main paper.

## B.1 PROOFS FOR SECTION 3.1 RESULTS NOT CONCERNING OPTIMAL POLICIES

We provide proofs for some of the results in section 3.1. The proofs relating to optimal policies (theorems 3.3 and 3.5) are given in the next subsection (appendix B.2).

**Theorem 3.1.** *Given an MDP and a policy $\pi$, the Q-function for $\pi$, $Q_\pi$, determines $R$ up to $S'$-redistribution. The optimal Q-function, $Q_\star$, has precisely the same invariances.*

*Proof.* $Q_\pi$ satisfies the Bellman equation for all $s \in \mathcal{S}$, $a \in \mathcal{A}$:

$$Q_\pi(s,a) = \mathbb{E}_{S' \sim \tau(s,a), A' \sim \pi(S')}\big[R(s,a,S') + \gamma \cdot Q_\pi(S',A')\big].$$

This equation can be rewritten as

$$\mathbb{E}_{S' \sim \tau(s,a)}\big[R(s,a,S')\big] = Q^\pi(s,a) - \gamma \cdot \mathbb{E}_{S' \sim \tau(s,a), A' \sim \pi(S')}\big[Q^\pi(S',A')\big].$$

From this, we can see that $Q^\pi$ is invariant to $S'$-redistribution of $R$, and no other transformations. $Q_\star = Q_{\pi_\star}$ where $\pi_\star$ is any optimal policy derived from $Q_\star$, so the invariances of the optimal $Q$-function follow as a special case. $\qquad\square$

**Theorem 3.2.** *Given an MDP, an inverse temperature parameter $\beta$, and a base policy $\pi_0$, the Boltzmann policy $\pi_\beta^{\pi_0}$ determines $R$ up to $S'$-redistribution and potential shaping. The Boltzmann-rational policy, $\pi_\beta^\star$, and the Maximum Entropy policy, $\pi_\beta$, have precisely the same invariances.*

*Proof.* By equation (2), $\pi_\beta^{\pi_0}$ can be derived from $A_{\pi_0}$. $A_{\pi_0}$ itself can be derived from $Q_{\pi_0}$, given $\pi_0$ (by equation (1), $A_{\pi_0}(s,a) = Q_{\pi_0}(s,a) - \mathbb{E}_{A \sim \pi_0(s)}\big[Q_{\pi_0}(s,A)\big]$). Thus $\pi_\beta^{\pi_0}$ is invariant to $S'$-redistribution by Theorem 3.1. Moreover, by Lemma A.1, potential shaping causes a state-dependent shift of $Q_{\pi_0}$. This shift does not affect $A_{\pi_0}$. Therefore $\pi_\beta^{\pi_0}$ is also invariant to potential shaping.

Conversely, recall (or see Lemma B.1, below) that the softmax function is invariant only to constant shifts. This means that $\pi_\beta^{\pi_0}$ is invariant to all and only those transformations of $R$ that produce state-dependent shifts in $Q^\pi$. Let $B : \mathcal{S} \to \mathbb{R}$, and suppose $R$ and $R'$ are two reward functions such that the corresponding $Q$-functions (for $\pi$) satisfy $Q'(s,a) = Q(s,a) + B(s)$. Then

$$
\begin{aligned}
\mathbb{E}[R'(s,a,S')] &= \mathbb{E}[Q'(s,a) - \gamma \cdot \max_{a' \in \mathcal{A}} Q'(S',a')] \\
&= \mathbb{E}[Q(s,a) + B(s) - \gamma \cdot \max_{a' \in \mathcal{A}}(Q(S',a') + B(S'))] \\
&= \mathbb{E}[B(s) - \gamma \cdot B(S') + R(s,a,S')]
\end{aligned}
$$

where the expectations are over $S' \sim \tau(s,a)$. Now set $\Phi(s) = -B(s)$, and we can see that the difference between $R$ and $R'$ is described by potential shaping and $S'$-redistribution.

The Boltzmann-rational policy determines its own base policy (it is enough to determine the optimal $Q$-function up to a state-dependent shift), and the Maximum Entropy policy *is* its own base policy, so the invariances of these policies follow as a special case of this result. $\qquad\square$

**Lemma B.1.** *Consider two functions $f : \mathcal{X} \to \mathbb{R}$ and $g : \mathcal{X} \to \mathbb{R}$ defined on a finite set $\mathcal{X}$. Then $g$ is constant if, for all $x \in \mathcal{X}$,*

$$\frac{\exp(f(x) + g(x))}{\sum_{x' \in \mathcal{X}} \exp(f(x') + g(x'))} = \frac{\exp(f(x))}{\sum_{x' \in \mathcal{X}} \exp(f(x'))}.$$

*Proof.* This is an elementary property of the softmax function. It can be seen as follows:

$$\to \qquad \frac{\exp(f(x) + g(x))}{\exp(f(x))} = \frac{\sum_{x' \in \mathcal{X}} \exp(f(x') + g(x'))}{\sum_{x' \in \mathcal{X}} \exp(f(x'))}$$

$$\to \qquad g(x) = \ln\left(\frac{\sum_{x' \in \mathcal{X}} \exp(f(x') + g(x'))}{\sum_{x' \in \mathcal{X}} \exp(f(x'))}\right)$$

which is constant in $x$. (The converse is true, but we do not make use of it, so we omit a proof.) $\quad\square$

**Theorem 3.4.** *Given an MDP, an inverse temperature parameter $\beta$, and a base policy $\pi_0$, the distribution of trajectories induced by the Boltzmann policy $\pi_\beta^{\pi_0}$ from all initial states determines $R$ up to $S'$-redistribution, potential shaping, and a mask of* unreachable *transitions. The distributions of trajectories induced by the Boltzmann-rational policy, $\pi_\beta^\star$, and the Maximum Entropy policy, $\pi_\beta$, from all initial states, have precisely the same invariances.*

*Proof.* That the distribution is invariant to $S'$-redistribution and potential shaping follows from Theorem 3.2. The distribution is also invariant to changes in the reward for transitions out of unreachable states, since these rewards cannot affect the policy for reachable states. As a result the distribution is additionally invariant to a mask of unreachable transitions.

The trajectory distribution can be factored into the separate distributions $\pi_\beta^{\pi_0}(s) \in \Delta(\mathcal{A})$ for each reachable state $s$, by conditioning on a supported prefix trajectory fragment that leads to $s$ and marginalising over subsequent states and actions. Via a similar argument to the proof of Theorem 3.2, the distribution determines the reward function for transitions (out of these reachable states) up to potential shaping and transformations and $S'$-redistribution (as they affect reachable states).

As for Theorem 3.2, the invariances for the Boltzmann-rational policy and the Maximum Entropy policy arise as special cases. $\qquad\square$

## B.2 Proofs for Section 3.1 Results Concerning Optimal Policies

Our results concerning the invariance of optimal policies and their trajectories follow from the following general result:

**Lemma B.2.** *Given an MDP $M$, suppose we have access to* the set of optimal actions *for each state in a* subset *of states $\mathfrak{S} \subseteq \mathcal{S}$. Assume that this subset is closed under optimal actions given the transition dynamics of $M$ (that is, taking optimal actions from states in $\mathfrak{S}$ never leads to states outside of $\mathfrak{S}$). Let $\mathfrak{D}$ be the set of functions $\mathcal{O}$ defined on $\mathcal{S}$ such that $\mathcal{O}(s) = \arg\max_{a \in \mathcal{A}} A_\star(s, a)$ for all $s \in \mathfrak{S}$ (but $\mathcal{O}$ is unconstrained for states outside $\mathfrak{S}$). Then these optimal action sets determine $R$ up to optimality-preserving transformations with $\mathcal{O} \in \mathfrak{D}$.*

*Proof.* Suppose $R'$ is obtained from $M$'s reward $R$ via an optimality-preserving transformation with some $\mathcal{O} \in \mathfrak{D}$. Let $\Psi$ be the corresponding value-bounding function, that is, a function $\Psi : \mathcal{S} \to \mathbb{R}$ satisfying, for all $s \in \mathcal{S}$ and $a \in \mathcal{A}$,

$$\mathbb{E}_{S' \sim \tau(s,a)}\big[R'(s, a, S') + \gamma \cdot \Psi(S')\big] \leqslant \Psi(s), \tag{3}$$

with equality if and only if $a \in \mathcal{O}(s)$. Since $\mathcal{O}(s)$ is non-empty (by definition), we have for all $s \in \mathcal{S}$

$$\Psi(s) = \max_{a \in \mathcal{A}} \big(\mathbb{E}_{S' \sim \tau(s,a)}\big[R'(s, a, S') + \gamma \cdot \Psi(S')\big]\big) \ .$$

This recursive condition on $\Psi$ is the Bellman optimality equation for the unique optimal value function, $V'_\star$, of the MDP with transformed reward $R'$. Therefore, $\Psi(s) = V'_\star(s)$ for all $s \in \mathcal{S}$, and we can rewrite equation (3) as

$$\mathbb{E}_{S' \sim \tau(s,a)}\big[R'(s, a, S') + \gamma \cdot V'_\star(S')\big] \leqslant V'_\star(s), \tag{4}$$

with equality only for $a \in \mathcal{O}(s)$.

Now, consider a state $s \in \mathfrak{S}$. By assumption, for this $s$, $\mathcal{O}(s) = \arg\max_{a \in \mathcal{A}} A_\star(s, a)$. Then for this state, the actions that attain the optimal value bound in equation (4) are these same optimal actions. Therefore, $R'$ induces the same sets of optimal actions from states in $\mathcal{S}$.

Conversely, consider a second MDP $M'$, differing from $M$ only in its reward function, $R'$. Assume the set of optimal actions in states in $\mathfrak{S}$ agrees with the optimal actions in $M$ for those states. Let $V'_\star$ and $A'_\star$ denote the optimal value and advantage functions for $M'$. The Bellman optimality equation for $M'$ ensures that, for $s \in \mathcal{S}$,

$$V'_\star(s) = \max_{a \in \mathcal{A}} \big(\mathbb{E}_{S' \sim \tau(s,a)}\big[R'(s, a, S') + \gamma \cdot V'_\star(S')\big]\big) \tag{5}$$

with the maximum attained precisely by the actions $a \in \arg\max_{a \in \mathcal{A}}(A'_\star(s, a))$. Setting $\mathcal{O}(s) = \arg\max_{a \in \mathcal{A}}(A'_\star(s, a))$, equation (5) can be rewritten as

$$\mathbb{E}_{S' \sim \tau(s,a)}\big[R'(s, a, S') + \gamma \cdot V_\star(S')\big] \leqslant V_\star(s) \tag{6}$$

for all $s \in \mathcal{S}$ and $a \in \mathcal{A}$, with equality if and only if $a \in \mathcal{O}(s)$.

Now, for $s \in \mathfrak{S}$, we have $\arg\max_{a \in \mathcal{A}}(A'_\star(s, a)) = \arg\max_{a \in \mathcal{A}}(A_\star(s, a))$, because $M$ and $M'$ have matching sets of optimal actions for these states (by assumption). Then equation (6) shows that $R'$ is produced from $R$ by an optimality-preserving transformation with $\mathcal{O}(s) = \arg\max_{a \in \mathcal{A}}(A'_\star(s, a))$ (and $\Psi(s) = V'_\star(s)$). $\qquad\square$

We are now in a position to prove the results from the main text:

**Theorem 3.3.** *Given an MDP, let $\mathcal{O}(s) = \arg\max_a A_\star(s, a)$. A maximally supportive optimal policy determines $R$ up to optimality-preserving transformations with $\mathcal{O}$.*

*Proof.* An arbitrary maximally supportive optimal policy determines the set of optimal actions from all states in the MDP. Its invariances follow as a special case of Lemma B.2, with $\mathfrak{S} = \mathcal{S}$. $\qquad\square$

**Theorem 3.5.** *Given an MDP, consider the distribution of trajectories induced by a maximally supportive optimal policy. Let $\mathfrak{S}$ be the set of states in supported trajectories. Let $\mathfrak{D}$ be the set of functions $\mathcal{O}$ defined on $\mathcal{S}$ such that $\mathcal{O}(s) = \arg\max_a A_\star(s, a)$ for all $s \in \mathfrak{S}$. The induced distribution of trajectories determines $R$ up to optimality-preserving transformations with $\mathcal{O} \in \mathfrak{D}$.*

*Proof.* The distribution of trajectories can be factored into separate distributions $\pi_\star(s) \in \Delta(\mathcal{A})$ for each state $s \in \mathfrak{S}$ (in a manner similar to Theorem 3.4, as proved above). These individual distributions determine the set of optimal actions within those states.

Noting that $\mathfrak{S}$ is clearly closed under optimal actions in the MPD since all optimal actions are supported by the policy, the invariance result follows from Lemma B.2. $\qquad\square$

## B.3 Proofs for Section 3.2 Results

**Theorem 3.6.** *Given an MDP, the return function restricted to possible trajectory fragments, $G_\zeta$, determines $R$ up to a mask of impossible transitions;*

*Proof.* The result is immediate, since the restricted domain still includes all possible transitions (as length one trajectory fragments with return equal to the reward of the transition), and no fragments with impossible transitions. $\qquad\square$

**Theorem 3.7.** *Given an MDP, the return function restricted to possible and initial trajectories, $G_\xi$, determines $R$ up to zero-initial potential shaping and a mask of unreachable transitions.*

*Proof.* The result follows from Lemma A.2 with $k = 0$. $\qquad\square$

**Theorem 3.8.** *Given an MDP, the distribution of comparisons of possible trajectory fragments, $\preceq^\zeta_\beta$, determines $R$ up to a mask of impossible transitions.*

*Proof.* Since $\preceq^\zeta_\beta$ can be derived from $G_\zeta$, it is invariant to a mask of impossible transitions by Theorem 3.6. Conversely, $\preceq^\zeta_\beta$ determines $R$ for all *possible* transitions. This is because $R(s, a, s')$ is encoded in the Boltzmann distribution of comparisons between the length zero trajectory fragment $\zeta_0 = (s)$ and the length one trajectory fragment $\zeta_1 = (s, a, s')$, and can be recovered as follows:

$$\mathbb{P}(\zeta_0 \preceq^\zeta_\beta \zeta_1) = \frac{\exp(\beta G(\zeta_1))}{\exp(\beta G(\zeta_0)) + \exp(\beta G(\zeta_1))} = \frac{\exp(\beta R(s, a, s'))}{\exp(\beta \cdot 0) + \exp(\beta R(s, a, s'))}$$

$$\rightarrow \qquad R(s, a, s') = \frac{1}{\beta} \cdot \ln\left(\frac{\mathbb{P}(\zeta_0 \preceq^\zeta_\beta \zeta_1)}{1 - \mathbb{P}(\zeta_0 \preceq^\zeta_\beta \zeta_1)}\right).$$

Therefore $\preceq^\zeta_\beta$ is invariant to precisely a mask of impossible transitions. $\qquad\square$

**Theorem 3.9.** *Given an MDP, the distribution of comparisons of possible and initial trajectories, $\preceq_\beta^\xi$, determines $R$ up to $k$-initial potential shaping and a mask of* unreachable *transitions.*

*Proof.* Note that as $\preceq_\beta^\xi$ can be derived from $G_\xi$, by Theorem 3.7, $\preceq_\beta^\xi$ is invariant to 0-initial potential shaping and a mask of unreachable transitions. It is additionally invariant to $k$-initial potential shaping for arbitrary constants $k \in \mathbb{R}$, and no other transformations: $G_\xi$ can be recovered from $\preceq_\beta^\xi$ up to a constant (we can compare all possible initial trajectories to an arbitrary reference trajectory and recover their relative return using a similar manipulation as above, but we can't determine the return of the reference trajectory). From there, the precise invariance follows from Lemma A.2. $\square$

**Theorem 3.10.** *We have the following bounds on the invariances of the noiseless order of possible trajectory fragments, $\preceq_\star^\zeta$. In all MDPs:*

(1) *$\preceq_\star^\zeta$ is invariant to positive linear scaling and a mask of impossible transitions; and*

(2) *$\preceq_\star^\zeta$ is not invariant to transformations other than zero-preserving monotonic transformations or masks of impossible transitions.*

*Moreover, there exist MDPs attaining each of these bounds.*

*Proof.* For (1), positive linear scaling of reward by a constant $c$ leads to the same scaling of the return of each trajectory fragment, and this always preserves the relation $\preceq_\star^\zeta$, since for any $c > 0$, $c \cdot G(\zeta_1) \leqslant c \cdot G(\zeta_2) \Leftrightarrow G(\zeta_1) \leqslant G(\zeta_2)$ for all pairs of trajectory fragments $\zeta_1, \zeta_2$. Moreover, $\preceq_\star^\zeta$ inherits invariance to a mask of impossible transitions from $G_\zeta$ (Theorem 3.6).

For (2), let $R'$ be produced from $R$ via some transformation that is *neither* a mask of impossible transitions *nor* a zero-preserving monotonic transformation. It must be that either $R'$ fails to preserve the ordinal comparison of two possible transitions, or that it fails to preserve the set of zero-reward possible transitions, compared to $R$. In the first case, consider two possible transitions whose rewards are not preserved, $x_1$ and $x_2$. Without loss of generality suppose $R(x_1) \leqslant R(x_2)$ but $R'(x_1) > R'(x_2)$. This corresponds to a change in $\preceq_\star^\zeta$'s comparison of the length one trajectories formed from $x_1$ and $x_2$, namely $x_1 \preceq_\star^\zeta x_2$ from true to false. Similarly, in the second case, the comparisons between the transition whose reward became or ceased to be zero and a length one trajectory (with return 0) will have changed. Therefore, $\preceq_\star^\zeta$ is not invariant to such transformations.

The bound (1) is attained by the following MDP invariant precisely to positive linear scaling and a mask of impossible transitions. Let $\mathcal{S} = \{s\}$, $\mathcal{A} = \{a_1, a_2\}$, $R(s, a_1, s) = 1$, and $R(s, a_2, s) = 1 + \gamma$. Since $R(s, a_2, s) = R(s, a_1, s) + \gamma R(s, a_1, s)$, the corresponding order relation will contain both $(s, a_2, s) \preceq_\star^\zeta (s, a_1, s, a_1, s)$ and $(s, a_1, s, a_1, s) \preceq_\star^\zeta (s, a_2, s)$. This property requires that $R(s, a_1, s) = (1 + \gamma) \cdot R(s, a_2, s)$, which is preserved only by linear scaling of $R$. (Non-positive linear scaling is already ruled out by (2)).

The bound (2) is attained by the following MDP invariant to arbitrary zero-preserving monotonic transformations. Let $\mathcal{S} = \{s_1, s_2\}$, $\mathcal{A} = \{a\}$, with possible transitions $(s_1, a, s_2)$ and $(s_2, a, s_2)$, and $R(s_1, a, s_2) > R(s_2, a, s_2) = 0$. Any zero-preserving monotonic transformation of $R$ preserves the ordering of all *possible* trajectory fragments, namely that all nonempty trajectories starting in $s_1$ have positive return and all other possible trajectories have zero return. $\square$

**Theorem 3.11.** *Given an MDP, the noiseless order of possible and initial trajectories, $\preceq_\star^\xi$, is invariant to $k$-initial potential shaping, positive linear scaling, and a mask of unreachable transitions.*

*Proof.* The pairwise Boltzmann distributions of $\preceq_\beta^\xi$ can be used to derive the noiseless comparisons of $\preceq_\star^\xi$. Therefore, $\preceq_\star^\xi$ is invariant to $k$-initial potential shaping and a mask of unreachable transitions.

That $\preceq_\star^\xi$ is also invariant to positive linear scaling follows from a similar argument as for the first bound in Theorem 3.10, proved above. $\square$

## B.4 Proofs for Section 3.3 Results

**Theorem 3.12.** *Given an MDP, let $\mathcal{O}(s) = \arg\max_a A_\star(s, a)$. Then the set of optimal policies determines $R$ up to optimality-preserving transformations with $\mathcal{O}$.*

*Proof.* The set of all optimal policies determines a maximally supportive policy, for example by constructing a policy that supports all actions supported by any policy in the set. Likewise, a maximally supportive policy determines the set of optimal policies, namely as the set of all policies whose support is a subset of the maximally supportive policy. Therefore, these objects share precisely the same invariances. $\square$

## B.5 Proofs for Section 4 Results

**Theorem 4.1.** *Consider an MDP $(\mathcal{S}, \mathcal{A}, \tau, \mu_0, R, \gamma)$, a policy $\pi$, and alternative transition dynamics $\tau'$ with $\tau(s, a) \neq \tau'(s, a)$ for all $s \in \mathcal{S}, a \in \mathcal{A}$. Given a function $Q' : \mathcal{S} \times \mathcal{A} \to \mathbb{R}$, there exists a reward function $R'$, produced from $R$ by $S'$-redistribution under $\tau$, such that $Q'$ is the Q-function for $\pi$ under $R'$ and $\tau'$.*

*Proof.* Per definition 2.3, that $R'$ is produced from $R$ by $S'$-redistribution under $\tau$ requires, for all $s \in \mathcal{S}$ and $a \in \mathcal{A}$,

$$\mathbb{E}_{S' \sim \tau(s,a)}\big[R'(s, a, S')\big] = \mathbb{E}_{S' \sim \tau(s,a)}\big[R(s, a, S')\big]. \tag{7}$$

Per equation (1), that $Q'$ is the Q-function for $\pi$ under $R'$ and $\tau'$ requires, for all $s \in \mathcal{S}$ and $a \in \mathcal{A}$,

$$Q'(s, a) = \mathbb{E}_{S' \sim \tau'(s,a), A' \sim \pi(S')}\big[R'(s, a, S') + \gamma Q'(S', A')\big],$$

or, equivalently,

$$\mathbb{E}_{S' \sim \tau'(s,a)}\big[R'(s, a, S')\big] = Q'(s, a) - \gamma \mathbb{E}_{S' \sim \tau'(s,a), A' \sim \pi(S')}\big[Q'(S', A')\big]. \tag{8}$$

For brevity, denote the right-hand sides of equations (7) and (8) as $\mathcal{R}(s, a)$ and $\mathcal{Q}(s, a)$ respectively.

Let $s \in \mathcal{S}$ and $a \in \mathcal{A}$. By assumption, $\tau'(s, a) \neq \tau(s, a)$. Therefore, for at least two $s'$, denoted $s'_1$ and $s'_2$, we have $\tau'(s'_1|s, a) < \tau(s'_1|s, a)$ and $\tau'(s'_2|s, a) > \tau(s'_2|s, a)$. For the remaining $s'$, set $R(s, a, s') = 0$. This reduces equations (7) and (8) to the following:

$$\tau(s'_1|s, a) \cdot R'(s, a, s'_1) + \tau(s'_2|s, a) \cdot R'(s, a, s'_2) = \mathcal{R}(s, a)$$
$$\tau'(s'_1|s, a) \cdot R'(s, a, s'_1) + \tau'(s'_2|s, a) \cdot R'(s, a, s'_2) = \mathcal{Q}(s, a)$$

This system has a solution as its determinant is $\tau'(s'_2|s, a) \cdot \tau(s'_1|s, a) - \tau(s'_2|s, a) \cdot \tau'(s'_1|s, a) > 0$.

$\square$

**Theorem 4.2.** *Given data sources $X$ and $Y$, let $(X, Y)$ denote the combined data source formed from $X$ and $Y$. If $X$ and $Y$ are incomparable, then $(X, Y) \prec X$ and $(X, Y) \prec Y$.*

*Proof.* Transformations that preserve $(X, Y)$ necessarily preserve $X$, therefore $(X, Y) \preceq X$. But since $X$ and $Y$ are incomparable, there is some transformation that preserves $X$ and not $Y$. This transformation does not preserve $(X, Y)$. Therefore, $(X, Y) \prec X$. Similarly, $(X, Y) \prec Y$. $\square$

We note that the above result is also an elementary consequence of the *lattice structure* of the partial order of partition refinement (Aigner, 1979, §I.2.B), since the combined data source corresponds to the *meet* of the original data sources.

# C    OTHER SPACES OF REWARD FUNCTIONS

Hitherto we have assumed reward functions are members of $\mathcal{S} \times \mathcal{A} \times \mathcal{S} \rightarrow \mathbb{R}$. That is, they are deterministic functions of transitions, depending on the state, action and the successor state. In this appendix, we discuss several alternative spaces of reward functions, and their implications for the invariance properties of various objects derived from the reward function.

## C.1    RESTRICTED-DOMAIN REWARD FUNCTIONS

It is common in both reinforcement learning and reward learning to consider less expressive spaces of reward functions. In particular, the domain of the reward function is often restricted to $\mathcal{S}$ or $\mathcal{S} \times \mathcal{A}$. When modelling a task, the choice of reward function domain is usually a formality: An MDP taking full advantage of the domain $\mathcal{S} \times \mathcal{A} \times \mathcal{S}$ has an 'equivalent' MDP with a restricted domain, and some added auxiliary states (Russell & Norvig, 2009, §17). Conversely, reward functions with restricted domains can be viewed as a special case of functions from $\mathcal{S} \times \mathcal{A} \times \mathcal{S}$ where the functions are constant in the final argument(s). Restricting the domain can be an appealing simplification when modelling a task, hence the popularity of these formulations.

When modelling a data source, this equivalence may not apply: We may not have access to data regarding auxiliary states, and so assuming a restricted domain effectively assumes the latent reward is indeed constant with respect to the successor state (and possibly the action) of each transition. This assumption may or may not be warranted.

If a restricted domain of $\mathcal{S}$ or $\mathcal{S} \times \mathcal{A}$ is preferred, then our invariance results can be adapted in a straightforward manner. In general, since we are effectively considering a subspace of candidate reward functions for transformations, ambiguity can only decrease. In particular, these restrictions have two main consequences.

Firstly, the reward function transformation of $S'$-redistribution vanishes to the identity transformation, since it allows variation only in the successor state argument of the reward function, which is now impossible. This reduces the effective ambiguity of the $Q$-function and all derivative objects. Notably, the $Q$-function uniquely identifies the reward function, and Boltzmann policies have the same invariances as Boltzmann comparisons between trajectories. Restricting the domain to $\mathcal{S}$ means the (state) value function for an arbitrary known policy also uniquely identifies the reward function, but doesn't otherwise alter the invariances we have explored.

Secondly, for most MDPs, the available potential shaping transformations are restricted, but not eliminated. The function added in a potential shaping transformation $(\gamma \cdot \Phi(s') - \Phi(s))$ nominally depends on the successor state of the transition. Some transformed reward functions may rely on this dependence, falling outside of the restricted domain. However, some non-zero transformations will usually remain. For example, in a discounted MDP without terminal states, a non-zero constant potential function $\Phi(s) = k$ does not *effectively* depend on $s$, and the reward transformation of adding $\gamma \cdot \Phi(s') - \Phi(s) = (\gamma - 1) \cdot k$ to a reward function does not introduce a dependence on $s'$. In general, the set of remaining potential shaping transformations will depend on the network structure of the MDP. At the extreme, in a deterministic MDP with state-action rewards, all potential shaping transformations are permitted, since a dependence on $s'$ can be satisfied by $a$.

## C.2    STOCHASTIC REWARD FUNCTIONS

Certain tasks are naturally modelled as providing rewards drawn stochastically from some distribution upon each transition. An even more expressive space of reward functions than we consider is the space of *transition-conditional reward distributions*[3]. Identifying the reward function in this case is more challenging in general because the latent parameter contains a full distribution of information for each input, rather than a single point. In the spirit of this paper, we sketch a characterisation of this additional ambiguity.

A deterministic reward function can be viewed as the conditional expectation of a reward distribution function. Taking the expectation of the reward distribution for each transition introduces invariance,

---

[3]Of course, it's also possible to consider reward to be distributed conditionally on only the state or state-action components of a transition, and not the full transition.

since the expectation operation is not injective (except in certain restricted cases such as for parametric families of distributions that can be parametrised by their mean). The invariance introduced is akin to $S'$-redistribution, but with an expectation over the support of the reward distribution rather than the successor state of each transition.

In the extension of the RL formalism to account for stochastic rewards, this expectation is effectively the first step in the derivation of each of the objects we have studied. Therefore, all of these objects inherit this new invariance.

As a consequence, all data sources are effectively more ambiguous with respect to this new latent parameter. For example, if optimal comparisons between trajectories are understood to be performed based on the pairwise comparison of the expected return of each individual trajectory, then these comparisons are also invariant to transformations of the reward distributions that preserve their means.

Fortunately, much of reinforcement learning also focuses on expected return and reward *in application*. Accordingly, most downstream tasks are tolerant to any ambiguity in the exact distribution of stochastic rewards, beyond identifying the mean. Since this is the same kind of ambiguity that is introduced by considering the latent parameter of reward learning as a conditional distribution rather than a deterministic function, our results are still informative for these situations.

### C.3 Further Spaces and Future Work

For certain applications, including *risk-sensitive RL* where non-mean objectives are pursued (Morimura et al., 2010a;b; Dabney et al., 2018), the distribution of stochastic rewards can be consequential. Moreover, the introduction of stochastic rewards suggests considering data sources based on samples rather than expectations, such as a data source of trajectory comparisons based on sampled trajectory returns. Characterising the invariances of these objectives to transformations of the reward distribution, and thereby their *ambiguity tolerance*, is left to future work.

In future extensions of this work to handle continuous MDPs, there will be an opportunity to study the effect of restricting to various parametrised spaces of reward functions. For example, it is common in reinforcement learning and reward learning to study MDPs with reward functions that are linear in a *feature vector* associated with each transition. This kind of restriction may reduce the available reward transformations compared to those available to a non-parametric reward function, in a similar manner to restricting the domain of a finite reward function as discussed above.

The relaxation of the Markovian assumption also introduces a broader space of reward functions, and with it, new dimensions for transformations and invariance. As one example, related to potential shaping, the non-Markovian additive transformations studied by Wiewiora et al. (2003) will amount to new invariances of the optimal policy and other related objects.

