# OpenReview forum: "Invariance in Policy Optimisation and Partial Identifiability in Reward Learning"
_ICLR.cc/2022/Conference — ICLR 2022 Submitted_

### Official Review · Reviewer_CDLH · 2021-11-02

**Correctness:** 4
**Technical Novelty And Significance:** 2
**Empirical Novelty And Significance:** Not applicable
**Recommendation:** 3
**Confidence:** 4

**Main Review:**

This is an unconventional paper. The problem it's attacking is very fundamental and interesting: given reward functions that are "close" to each other under some measure (doesn't have to be rigorous and literal "measures"), what will they affect (in both the optimum of the MDP and the behavior of the downstream algorithms). What the paper does is to provide a set of measures and a set of claims that partially answer the questions, for the cases that are more or less low-hanging fruits. These claims are not particularly strong I would say, but they do provide some reasoning and implications on this very important topic. I would personally be more interestied in a particular, presumably not that general, setting (say, just tabular and linear program), with a clear and complete answer to that question. To this end I would wonder if the manuscript better fits a journal publication or a book chapter etc. But I'm more or less open for discussion in case this manuscript could provide some support for future studies.

Side question: Can authors just compile one single pdf of 21 pages for the main submission per the policy?

**Summary Of The Paper:**

This paper characterizes the partial identifiability of data sources and the reward function. Then it analyzes the impact of this measure on the optimum and the algorithms. Some implications are given.

**Summary Of The Review:**

Better fits journal publication or book chapter?

---

> ### Author Response · Authors · 2021-11-23
> **Author Response to Official Review of Reviewer CDLH**
>
> Thank you for your review and for your suggestions. We especially appreciate your recognition of the topic of the paper as important, fundamental, and interesting.
>
> **Regarding the conventionality of the paper:** While we admit that our work is theoretical in nature, we don't see the work as 'unconventional' for ICLR. ICLR has accepted papers on a broad range of topics, and explicitly lists "theoretical issues in deep learning" and "reinforcement learning" in the [call for papers](https://iclr.cc/Conferences/2022/CallForPapers). Could you elaborate on how you see the paper as being unconventional, and in particular any results you would like to see added?
>
> **Regarding the preference for a detailed analysis of a special case:** We thank you for your suggestion towards extending the paper. We note that reviewer **zcgk** has proposed a similar extension.
>
> Regrettably, we are not in a position to include such a case-study with this paper submission. We note that many of our results offer precise bounds on the class of invariances, and as such, they do form 'complete answers' for a wide range of MDPs. The results might not be immediately 'clear' due to their generality, but the concretisation to any specific MDP is simply an instantiation of the results---we think there is little additional work to carry out by the practitioner who wants results for their specific MDP.
>
> Otherwise, as you have noted, we have attempted to draw out 'reasoning and implications' of the results rather than merely presenting a list of abstract results.
>
> **Regarding support for future work:** We're not quite sure what you had in mind as providing 'support for future studies'. We note that we view this paper as a first step towards a rich body of theory for the statistical evaluation of different reward learning problems. However, if this is not what you had in mind, we would appreciate it if you could clarify your suggestion.
>
> There are many directions for future work. Some important directions are mentioned in the paper. One direction that is not mentioned in the paper, but that is perhaps relevant to this discussion, is that one can build upon the framework and concepts established by this paper in studying the invariance properties of a wider range of reward learning data sources, including more realistic models of human behaviour as appropriate for real-world reward learning. Our work attempts to lay out the case for this kind of analysis, and a conceptual and mathematical framework that is much more broadly applicable than in our results (e.g. the partial order of data sources and applications by ambiguity (tolerance) refinement) and is therefore a crucial first contribution in this direction.
>
> **Regarding novelty:** We noticed that you rated the contributions as 'only marginally significant or novel'. We would respectfully suggest that the paper makes a substantial novel theoretical and conceptual contribution.
>
> Each of our theorems extend existing results, and together in number they would appear to represent a major extension of existing theoretical analysis in this area. This is true for the results regarding inverse reinforcement learning, which generalise and extend existing results in the reward learning literature as discussed in the related work section. It's even more the case for the results concerning preference comparisons, which have not been formalised in the reward learning literature (or, to our knowledge, in any other literature).
> Moreover, we view the conceptual framework (including the unified study of ambiguity and tolerance using the partial order of ambiguity tolerance refinement) as significant and novel. For what it's worth, reviewer **iFN9** agreed that "this work makes substantial contributions by conducting this study in a unified and rigorous way for variety data sources and downstream tasks."
>
> **Journal publication:** We are flattered by your suggestion that we should publish this work in a journal or book chapter. We agree that the more generous page limit allowed by these venues would allow us to provide a more detailed exposition of our theoretical framework. However, we think a short conference version will be a useful summary for practitioners who wish to understand our headline results, without needing to dive into the details of the derivation. We are not sure how to reconcile the recommendation for journal publication with your score -- although ICLR is admittedly a competitive venue, so too are journals such as JMLR or JAIR.
>
> ---
>
> Once again, we would like to thank you for your review and suggestions. We hope that our response might earn your reconsideration of the score you have awarded.

---

### Official Review · Reviewer_zVCQ · 2021-11-07

**Correctness:** 2
**Technical Novelty And Significance:** 2
**Empirical Novelty And Significance:** Not applicable
**Recommendation:** 3
**Confidence:** 1

**Main Review:**

This paper is very hard to follow due to many non-standard notation and concepts. It is not even clear what the goal of the paper is. The notion of invariance is vague and confusing. I think the authors should have given more effort to setup the problem and goal more precisely and concisely, while using easier to understand notation and terminologies.

**Summary Of The Paper:**

The submission considers a reward learning problem when the reward function is not uniquely recoverable from the data, even in the infinite-data regime.

**Summary Of The Review:**

The paper is very hard to follow and I may not be able to assess the paper properly.

---

> ### Author Response · Authors · 2021-11-23
> **Author Response to Official Review of Reviewer zVCQ**
>
> Thank you kindly for your review and for your suggestions. We are sorry to hear that you found the paper difficult to follow. We are definitely interested in maximising the accessibility, clarity, and readability of the paper. We also appreciate your arranging for an additional reviewer to take another look at our paper.
>
> **Regarding the goal of the paper:** We are sorry to hear that you found the motivation and goal of the paper unclear. For what it's worth, we can state the goal as follows: *We present a unified framework for studying the fundamental limits of information recoverable about reward functions from different data sources, and the robustness of applications (including policy optimisation in RL) to these limits.*
>
> We essentially stand by our opening section, which, occupying just over one half of one page, we have made as concise and accessible as we could manage, while addressing the two core aspects of our analysis (the partial identifiability of reward learning models, and the tolerance of applications) and their duality, leading to the unified framework, which we see as our conceptual contribution.
>
> **Regarding the use of difficult and non-standard notation:** We have adhered to standard notation, terminology, and concepts where possible. Regrettably, as pointed out by reviewer **iFN9**, it is inevitable that some new notation is introduced when discussing novel concepts. We have been careful to make consistent use of terminology throughout the paper.
>
> Do you have any suggestions for alternative notation or structure of the paper that you would have found easier to read?
>
> **Regarding the specific concept of invariance:** We would like to note that invariance is a well-studied concept in mathematics, which we believe we are using in an established and standard sense. We admit that our definition of invariances as a mathematical concept omits some mathematical details in favour of a more compact presentation. We also have attempted to provide some alternative, intuitive explanations of the use of the concept, including the relation to the equivalence kernel of the derivation function.
>
> ---
>
> Once again thank you for taking the time to review our work and for your suggestions that there is room for improvement in readability. If you are able to list any points of particular difficulty, we would be happy to address them for the camera-ready submission if our paper is eventually accepted.

---

### Official Review · Reviewer_iFN9 · 2021-11-08

**Correctness:** 4
**Technical Novelty And Significance:** 3
**Empirical Novelty And Significance:** Not applicable
**Recommendation:** 8
**Confidence:** 4

**Main Review:**

The paper is overall well written. The related work is clearly discussed, and this work is well-positioned in the literature. I enjoyed reading the paper.

I haven’t checked the proofs of the theorems; however, the justifications provided before the theoretical claims are convincing.

The paper is a bit heavy in terminologies; however, it is inevitable due to the theoretical nature and rigor purpose of the paper.

Comments:
1/ Re. the optimality-preserving transformations in Definition 2.5: is it possible to extend this definition to regularized MDP (e.g., entropy regularized MDP) objective as well? In that case, will the function \Psi intuitively correspond to the regularized value function?

2/ In Theorem 3.3, the S’-redistribution invariance and potential shaping invariances are not mentioned -- is this because they are less ambiguous than optimality-preserving transformations?

3/ Re Theorem 3.10 for noiseless comparison based on the return: compared to other objects/theoretical statements, in this theorem, there isn't any “determines”-style claim. It is said that the precise monotonic invariances depend on the MDP. It would be good to have a clarification/discussion on this point.

4/ All the results are derived for the finite MDP setting. A brief discussion on the applicability (or possible extension) of these claims to the continuous MDP setting would be helpful.


**Summary Of The Paper:**

This is a theoretical work on understanding the intrinsic limits of various data sources that are used for reward learning in RL. In particular, by considering the infinite data limit of the data source, they study the level of reward ambiguity that can be obtained for a given downstream task. For example, for the expert behavior data source, they characterize the reward transformations that are determined by the optimal Q-function. Similar attempts have been made previously for specific data sources and specific planning algorithms (Ng & Russell, 2000). However, this work makes substantial contributions by conducting this study in a unified and rigorous way for variety data sources and downstream tasks.

**Summary Of The Review:**

This is a strong theoretical work; it makes a fundamental contribution to reinforcement learning literature. The reward functions are atomic to RL -- understanding the theoretical limits on how much information can be extracted from various data sources used for reward learning is important.

---

> ### Author Response · Authors · 2021-11-23
> **Author response**
>
> We thank you for your detailed review and suggestions. We especially appreciate your recognition that the unified framework is a substantial contribution and that the problem we have studied is of fundamental importance to reinforcement learning.
>
> We offer the following responses to each of your comments. We also plan to address some of these in minor modifications to the manuscript in an updated version that will be uploaded in time for today's paper submission deadline.
>
> **1. Regarding optimality-preserving transformations for entropy-regularised MDPs:**
> We haven't explored this direction, however we speculate that the definition will generalise trivially through incorporation of the soft value function into psi, as you suggest. In any case, an entropy-regularised policy will have the form of a Boltzmann distribution, which offers more information than an argmax-based policy (namely relative information about the value of suboptimal actions) and thus general optimality-preserving transformations will not be necessary to describe their invariance.
>
> **2. Regarding theorem 3.3:** Yes, it is exactly as you suggest, and this is alluded to in the paragraph before theorem 3.3 and made more explicit in section 4.
>
> **3. Regarding theorem 3.10:** Theorems 3.10 and 3.11 indeed use different wording than the other results. Theorem 3.11 is a one-sided bound on the invariances: The object is invariant to these transformations, and possibly others that are not combinations of these transformations. Theorem 3.10 is a little subtler since depending on the structure of the MDP, the objects may or may not be invariant to certain transformations. However, the bounds are as tight as possible for general MDPs, as discussed in and before the theorem statement.
>
> If we understand correctly, you suggest a clarification and discussion of these points. In today's updated paper submission, we will add a reference to the proof in the appendix which discusses specific examples of MDPs at either extreme (as part of the proof that these bounds are tight). If the paper is accepted, then for a camera-ready submission, we would have space to add a brief note on some general features of MDPs that allow or restrict these invariances (for example based on the constructions in the proof that these bounds are tight).
>
> **4. Regarding the possible extension of these results to continuous MDPs:**
> Thank you for your suggestion. We have yet to explore this direction in detail, leaving it to future work. We expect that the framework for our analysis will be applicable without modification, as continuous MDPs merely offer an expanded space of reward functions relative to finite MDPs. It is therefore still possible to discuss transformations, invariances, and the partial order by ambiguity (tolerance) refinement of objects based on this space. However, the details of the results will depend on the exact allowed space of reward functions: for example, unrestricted continuous reward functions, or a parametric family such as linear reward functions.
>
> Due to space constraints in this conference submission, we decided not to dedicate space in the main paper towards this discussion. However, we do include a brief comment on different spaces of reward functions including for continuous MDPs in appendix C (viz. second paragraph of section C.3). We would be happy to elaborate on this in the appendix as part of a camera-ready submission if the paper is accepted.
>
> ---
>
> Once again, we would like to thank you for your detailed review and suggestions. We hope that our responses to your questions are helpful.

---

### Official Review · Reviewer_zcgk · 2021-11-16

**Correctness:** 4
**Technical Novelty And Significance:** 3
**Empirical Novelty And Significance:** Not applicable
**Recommendation:** 8
**Confidence:** 4

**Main Review:**

Some clarification questions:

1. At the end of page 3, you mentioned the maximum entropy policy as the fixed point to $\pi_\beta = \pi_{\beta}^{\pi_\beta}$. Could you provide a definition of the maximum entropy policy and a proof sketch of why it is equivalent to the fixed point? In my understanding, a maximum entropy policy is the optimal policy result from solving policy optimization with an entropy regularization. It's not obvious to me why this solution will coincide with the fixed point defined above.

2. X never appears in Definition 2.1. Do you mean to define $X = f(R) = f(t(R))$?

3. I don't quite understand Definition 2.5, as the mathematical definition of $R'$ seems to be independent of $R$?

4. 5 transformations are defined in section 2.1. Do they mean to form an exhaustive list? Or are there potentially other relevant transformations not discussed here? At a higher level, an object (e.g. expert demonstration) can induce a set of reward functions not described by any finite combination of the listed or potentially new transformations. So instead of characterizing the information flow through the reward function, why not directly analyze whether the information from a data source is sufficient for a downstream task? The answer to this question is completely independent of whether one chooses to learn a reward function or not. For example, the information captured in an expert demonstration is sufficient for the downstream task of optimal policy identification, and therefore it is well-known that despite the non-uniqueness of reward function in IRL, an optimal policy can be recovered regardless.

Major comments:

**Discussion of related works can be significantly improved**: Most of the mentioned related works on reward learning are empirical in nature. It would be good to instead comprehensively survey the theoretical works that concern reward learning, since the paper itself is theoretical in nature.
**Example application of the framework to improve upon/subsume prior results**: While the introduced reward-function centered information-theoretical framework provide a novel and unified view on the information-theoretical relationship between different objects, it alone is in my opinion not sufficient for publication. When you introduce a brand-new framework, it is also important that you make some convincing argument of why this framework is **useful**, in addition to being mathematically correct. A typical way of doing this is to show that using the new framework, one can easily recover/improve results from various prior works, and thus truly provide a unifying framework that subsumes prior literature.

I would suggest the authors throw a majority of the theorems and transformations in section 2 and section 3 into the appendix, and use the space to set up one or two concrete examples on which the proposed framework can be useful in deriving new/matching upper and lower bounds.


**Summary Of The Paper:**

This paper provides a theoretical framework to understand the relationship between data source/downstream tasks and reward functions.

**Summary Of The Review:**

1. Discussion of related works can be significantly improved.
2. Example application of the framework to improve upon/subsume prior results.

In summary, I think this paper provides some novel insight in the problem of reward learning, but there can be substantial improvements to be made to make the paper significantly stronger. I would highly suggest the authors make the additional effort, and I know it's gonna be a lot of work. But it will potentially make it a spotlight/oral paper instead of a borderline.

---

> ### Author Response · Authors · 2021-11-23
> **Author response (Part 1---Major Comments)**
>
> **In response to your first major comment, regarding related work:**
> We sympathize with your concern that most of the related work cited is empirical in nature. Unfortunately, we are not aware of any additional related theoretical work. Do you know of any? If so, we would be happy to incorporate it.
>
> We agree that the discussion of the theoretical work could benefit from more detailed elaboration. In particular, the existing results of Ng & Russell (2000), Dvijotham & Todorov (2010), Amin et al. (2017), Kim et al. (2021), and Cao et al. (2021). However, we felt it would be inappropriate to unpack these results in our early related work section due to the dense mathematical nature of the results. We have made a very brief mention of the relation between our results and those of Amin et al. (2017) and Cao et al. (2021) at the end of section 4, but we did not have space to elaborate further in a short conference paper.
>
> Although our work is theoretical, we felt it was important to discuss empirical work as well. In particular, we want to cite a breadth of work in the general field of "reward learning" to help contextualise our work, and the majority of existing work is empirically focused. Moreover, the empirical focus of most prior work serves to motivate our theoretical contribution, which is so far relatively lacking in this field. We'd like to point out that reviewer **iFN9** found the related work section made our work "well-positioned in the literature".
>
> We wonder if these considerations affect your judgement of the related work section as requiring significant improvement.
>
> ---
>
> **In response to your second major comment, regarding demonstrating the usefulness of the framework:**
> We thank you for your suggestion to extend the paper with a detailed case-study of the framework in a concrete application. We note that reviewer **CDLH** has suggested a similar extension.
>
> Regrettably, we are not in a position to include such a case-study with this paper submission. We share the goal of justifying the framework's usefulness and we view this as part of the contribution of section 4 ("Implications for reward learning"). Otherwise, we note that many of our results offer precise bounds on the class of invariances, and as such their application to any MDP is simply an instantiation of the results---we think there is little additional work to carry out by the practitioner who wants results for their specific MDP.

---

> ### Author Response · Authors · 2021-11-23
> **Author response (Part 2---Clarifying questions 1 and 2)**
>
> **1. Regarding the fixed-point definition of the maximum entropy policy:**
> Thank you for drawing our attention to this issue. Upon revisiting Haarnoja et al. (2017), we found that this relationship is not stated as we had thought it was. We agree that this is a significant issue with the current submission, and we hope to be able to address it to make the paper robust. However, we are confident that the claimed conclusions are correct with respect to the more standard entropy regularisation-based definition of the maximum entropy policy.
>
> We suspect that the fixed point criterion follows from the entropy regularisation definition of the maximum entropy policy, which is enough to establish our results, since all they assume is the fixed-point criterion. A proof sketch is as follows:
>
> Equation (4) of Haarnoja et al. (2017) decomposes into a hard Q function of the maximum entropy policy plus a state-based expected-future-entropy function. Then, when taking the Boltzmann distribution of the soft Q value to get the maximum entropy policy (equation (3) of Haarnoja et al.), the state-based additive expected-future-entropy contribution cancels due to normalisation. Thus, in our terminology, the maximum entropy policy is the Boltzmann policy with respect to its own (hard) Q function (as well as its own soft Q function); this is the fixed point criterion we require.
>
> We will need more time (until the camera-ready submission) to clarify this relationship in the paper, by which point we could include this derivation in an appendix and a note along with its introduction.
>
> We think this sketch will lead to a correct proof. However, we need more time to check it carefully. As a backup, we have an alternative sketch for why, even if we have to discard the fixed point criterion and start from the entropy regularisation based definition of the maximum entropy policy, all of the claimed results still follow (i.e. our conclusions do not depend fundamentally on the fixed point criterion):
>
> The relevant part of theorem 3.2 follows via similar reasoning as for the proof of theorem 3.2. By equation (6) in Haarnoja et al. (2017), the entropy regularisation based maximum entropy policy determines the soft Q function up to a state-based constant; which itself determines the reward function up to potential shaping and S'-redistribution.
>
> **2. Regarding X in definition 2.1:** We will adopt your suggestion (X = f(R) = f(t(R))), thanks, we agree that this clarifies the definition.

---

> ### Author Response · Authors · 2021-11-23
> **Author response (Part 3---Clarifying questions 3 and 4)**
>
> **3. Regarding dependence of definition 2.5 on R:** This definition is a little different from some of the others, in that the dependence on R is implicit. In use (e.g. thms 3.5, 3.12, appendix B.2), we consider particular $\mathcal{O}$ functions, and the dependence on R arises implicitly through the definition of these $\mathcal{O}$ functions. For example, in thm 3.5, we let $\mathcal{O}$ be the argmax of the optimal advantage function under R, which depends on R.
> We will reword the explanatory note under definition 2.5 to point out this dependency. Thank you for bringing this issue to our attention.
>
> **4. Regarding the set of transformations:** These transformations are not intended to be exhaustive. We defined transformations sufficient to express the results for the classes of transformations we study. These transformations appear to be independently interesting, however objects of future investigation may necessitate the definition and study of new classes of transformations.
>
> **Regarding the choice to make everything relative to the reward function:** We agree with your observation that one could analyse the relation between objects directly, without comparing them to the reward function. We see this as an alternative perspective on our results, with more emphasis on the local structure of the relations between objects (e.g. figure 1a). This 'algebra of reinforcement learning' perspective reveals new, more general motivations for the work.
>
> For this conference presentation, we have chosen to focus on a single perspective, and we have chosen the reward-centric perspective in particular, in an attempt to appeal to researchers in the field of reward learning. Reward learning appears to have the most immediate need for theoretical analysis in the face of partial identifiability. Moreover, reward functions seem to offer an elegant organising principle for these results, since they are of central importance in reward and reinforcement learning (as also remarked by reviewer iFN9), and they allow us to tell a unified story following the step-wise derivation of our diverse objects from their roots in reward functions.
>
> Moreover, we think that it is still quite possible to take an alternative reading of our results and proofs in line with this more abstract 'algebra of reinforcement learning' perspective, i.e. we have not lost anything technical by committing the text to a reward learning perspective. For instance, many of our proofs proceed by establishing the invariances to the reward function via invariances to transformations affecting some intermediate object. These same arguments can be refactored into proofs of the invariance relationship between specific pairs of non-reward objects.

---

> ### Author Response · Authors · 2021-11-23
> **Author response (Overview)**
>
> Thank you for your detailed review and for your suggestions. We are especially grateful that you have taken the time to review our paper as an additional reviewer after one of the first three reviews expressed low confidence in their review. We will reply to your two major comments and your four clarifying questions in the following three messages.
>
> In response to your clarifying questions, we are making some adjustments to the manuscript which will be submitted by the paper submission deadline, and we have the replies below.
>
> Once again, we would like to thank you for your detailed review and suggestions. We hope that our responses below, especially regarding the clarifying questions and the related work section, might earn your reconsideration of the borderline score you have awarded.

---

### Author Response · Authors · 2021-11-23
**Minor revision to paper and abstract**

We have submitted a minor rebuttal revision ahead of tonight's paper submission deadline. The revised paper includes a few extra words of additional detail in response to some suggestions and questions from our reviewers as discussed in the review threads below. We have also made a small number of miscellaneous minor improvements to the expression of a few paragraphs. None of the definitions or theorems have been altered and no substantially new sections have been added (nor even new paragraphs), as can be verified in the diff.

**Abstract:** Perhaps the most noticeable change is to the abstract, which has been revised as follows:

> Designing reward functions for complex, real-world tasks is challenging.
*Reward learning* lets one instead *infer* reward functions from data.
However, multiple reward functions often fit the data equally well, even in the infinite-data limit.
Prior work often considers reward functions to be uniquely recoverable, by imposing
additional assumptions on data sources.
By contrast, we formally characterise the *partial identifiability*
of popular data sources, including demonstrations and trajectory preferences, under
multiple standard sets of assumptions.
We analyse the impact of this partial identifiability on downstream tasks such as
policy optimisation, including under shifts in environment dynamics.
We unify our results in a framework for comparing data sources and downstream tasks
by their invariances,
with implications for the design and selection of data sources for reward learning.

---

### Decision · Program_Chairs · 2022-01-20

**Decision:**

Reject

**Comment:**

The paper formally studies the problem of partial identifiability when inferring a reward function from a given data source (e.g., expert demonstrations or trajectory preferences). To formally characterize this ambiguity in a data source, the paper proposes considering the infinite-limit data regime, which bounds the reward information recoverable from a source. Furthermore, this ambiguity is then studied in the context of different downstream tasks, as recovering an exact reward function may not be necessary for a given task. The paper is primarily theoretical, and the results provide a unified view of the problem of partial identifiability in reward learning for different sources and downstream tasks.

Overall, the reviewers acknowledged the importance of the problem setting and found the results promising. There is quite a bit of spread in the reviewers' final assessment of the paper with ratings 8, 8, 3, 3 (note: one of the reviewers with rating 3 has a low confidence). The authors' responses did help in discussions; however, a few of the concerns, as raised by reviewers, still remained. The key issues are related to the general accessibility of the paper and the lack of concrete examples to highlight the proposed theoretical framework. At the end of the discussions,  several reviewers (including those with an overall positive rating) shared concerns about the paper's accessibility.  With this, unfortunately, the paper stands as borderline. Nevertheless, this is exciting and potentially impactful work, and we encourage the authors to incorporate the reviewers' feedback when preparing a future revision of the paper.